# Acceleration of Neutral Atoms by Strong Short-Wavelength Short-Range Electromagnetic Pulses

Vladimir S. Melezhik [1,2,*] and Sara Shadmehri [1,*]

1    Bogoliubov Laboratory of Theoretical Physics, Joint Institute for Nuclear Research,
     Dubna 141980, Moscow Region, Russia
2    Dubna State University, 19 Universitetskaya Street, Dubna 141982, Moscow Region, Russia
*    Correspondence: melezhik@theor.jinr.ru (V.S.M.); shadmehri@theor.jinr.ru (S.S.)

**Abstract:** Nondipole terms in the atom–laser interaction arising due to the presence of a magnetic component in an electromagnetic wave and its inhomogeneity lead to the nonseparability of the center-of-mass (CM) and electron variables in the neutral atom and, as a consequence, to its acceleration. We investigate this effect and the accompanying excitation and ionization processes for the hydrogen atom in strong ($10^{12} - 2 \times 10^{14}$ W/cm$^2$) linearly polarized short-wavelength (5 eV $\lesssim \hbar\omega \lesssim$ 27 eV) electromagnetic pulses of about 8 fs duration. The study was carried out within the framework of a hybrid quantum-quasiclassical approach in which the coupled time-dependent Schrödinger equation for an electron and the classical Hamilton equations for the CM of an atom were simultaneously integrated. Optimal conditions with respect to the frequency and intensity of the electromagnetic wave for the acceleration of atoms without their noticeable ionization were found in the analyzed region.

**Keywords:** nondipole effects; atomic acceleration; quantum-quasiclassical approach; strong laser field

## 1. Introduction

The influence of nondipole effects $\sim 1/c$ (here, $c = 1/\alpha = 137$ is the speed of light in the atomic system of units (a.u.)) on various atomic processes in strong laser fields arising due to the presence of a magnetic component in the laser field and its inhomogeneity is currently being intensively and widely investigated (see, for example, [1–14], and the references therein). In particular, their influence on the "stabilization" of atoms at high laser intensities (the probability of ionization reaching a plateau significantly below unity) [9] and the generation of high harmonics (even harmonics, which are forbidden in the dipole approximation, appear in the atomic emission spectrum) [14] are predicted. However, nondipole effects, leading to nonseparability of the center-of-mass (CM) and electron variables in an atom interacting with a laser pulse, remain underexplored [15]. We believe this is due to the computational complexity of the problem at hand. Even in the simplest case, in the problem of a hydrogen atom interacting with laser radiation, taking into account these terms, which "entangle" the electron and proton variables in the Hamiltonian of the problem, leads to the need to solve the six-dimensional time-dependent Schrödinger equation.

Nevertheless, the problem of the hydrogen atom in a strong laser field ($10^{14}$ W/cm$^2$) taking into account the motion of the proton (due to the nondipole effect of the nonseparability of the CM in this case) was investigated within the framework of the quantum-quasiclassical method in Melezhik's recent work [16]. In this approach, the electron is treated quantum mechanically and the CM motion classically. Thus, the Schrödinger equation for the electron and the classical Hamilton equations for the CM variables, which are nonseparable due to nondipole effects stimulated by strong laser fields, are integrated simultaneously. In particular, it was shown that, with an increase in photon energy from 1.5 eV to 13.6 eV, the hydrogen atom can be accelerated to a velocity of $\sim$2 m/s at an intensity of $10^{14}$ W/cm$^2$ and a pulse duration of $\sim$2 fs, which does not contradict the available

experimental results [1], where it was possible to accelerate helium and neon atoms to a velocity of $\sim 50$ m/s in femtosecond laser pulses of intensity $8 \times 10^{15}$ W/cm$^2$ with a photon energy of 1.0–1.5 eV. The problem is attractive due to new possible applications in both fundamental and applied physics. Therefore, some evaluations of atom acceleration using strong laser fields have already been performed (see, for example, [15,17–19]).

In this work, we explore the possibility of accelerating a hydrogen atom, and its excitation and ionization using strong ($10^{12} - 2 \times 10^{14}$ W/cm$^2$) linearly polarized short-wavelength (5 eV $\lesssim \hbar\omega \lesssim 27$ eV) electromagnetic pulses of about 8 fs duration. The study of atoms in strong laser fields lasting several tens of fs in this intensity range is currently the subject of intensive experimental research (see, for example, [7,17,20] and the refs. therein). The intensity and pulse duration region of about $10^{14}$ W/cm$^2$ and a few fs is also typical for theoretical investigations (see, for example, [8,13,15,21,22]). The analysis is carried out with a fairly small photon energy step in the region that has a resonance character near the ionization threshold. This may be required for choosing optimal conditions when planning experiments. Our investigation was performed in the framework of the quantum-quasiclassical approach [16], the main elements of which are described in Section 2. In Section 3, we present the results of the nondipole calculations of the acceleration, excitation, and ionization of the hydrogen atom using high-intensity laser fields. In the considered range of laser field parameters (intensity and frequency), optimal conditions for atomic acceleration were found. It is also shown that the influence of nondipole effects in the considered range of laser intensities and frequencies on the values of ionization and excitation of the atom is insignificant. The last section is devoted to a short conclusion.

## 2. Theoretical Method

We have studied the dynamics of a hydrogen atom in a strong laser field linearly polarized along the $z$-axis and propagating along the $y$-axis. Assuming a sine-squared carrier envelope for the laser pulse, the vector potential of the laser field is given by

$$\mathbf{A}(\mathbf{r}, t) = \hat{\mathbf{z}} \frac{E_0}{\omega} \sin^2\left(\frac{\pi t}{NT}\right) \sin(\omega t - \mathbf{k} \cdot \mathbf{r}), \tag{1}$$

where $E_0$ is the strength of the field defined by the field intensity $I = \epsilon_0 c E_0^2 / 2$ ($\epsilon_0$ is the vacuum permittivity), $\omega$ is the frequency of the laser field, and $\mathbf{k} = k\hat{\mathbf{y}} = \omega/c\hat{\mathbf{y}}$ and $c$ are the wave vector and the speed of light, respectively. Here, $N$ shows the number of optical cycles in the period $T = 2\pi/\omega$, which are included in the laser pulse.

By going beyond the dipole approximation and expanding the space dependent $\mathbf{A}$ to the first order of $\omega y / c$, the vector potential $\mathbf{A}$, and the electric $\mathbf{E} = -\frac{d\mathbf{A}}{dt}$ and magnetic $\mathbf{B} = \nabla \times \mathbf{A}$ fields of the laser pulse take the forms

$$\mathbf{A}(\mathbf{r}, t) = \hat{\mathbf{z}} \frac{E_0}{\omega} \sin^2\left(\frac{\pi t}{NT}\right) \left[ \sin(\omega t) - \frac{\omega}{c} y \cos(\omega t) \right], \tag{2}$$

$$\begin{aligned} \mathbf{E}(\mathbf{r}, t) = \quad &- \quad \hat{\mathbf{z}} E_0 \sin^2\left(\frac{\pi t}{NT}\right) \left[ \cos(\omega t) + \frac{\omega}{c} y \sin(\omega t) \right] \\ &- \quad \hat{\mathbf{z}} E_0 \frac{1}{2N} \sin\left(\frac{2\pi t}{NT}\right) \left[ \sin(\omega t) - \frac{\omega}{c} y \cos(\omega t) \right], \end{aligned} \tag{3}$$

$$\mathbf{B}(\mathbf{r}, t) = -\hat{\mathbf{x}} \frac{E_0}{c} \sin^2\left(\frac{\pi t}{NT}\right) \cos(\omega t). \tag{4}$$

The Lorentz force acting on each constituent particle of the hydrogen atom (with charge $q_i = \pm e$, mass $m_i$, and momentum $\mathbf{p}_i$) is of the form $\mathbf{F}_i = q_i \mathbf{E} + \frac{q_i}{m_i}(\mathbf{p}_i \times \mathbf{B})$, where $i = 1, 2$ denotes the electron and proton, respectively. Passing to the coordinates of CM

($\mathbf{R} = (m_1\mathbf{r}_1 + m_2\mathbf{r}_2)/M$, $\mathbf{P} = \mathbf{p}_1 + \mathbf{p}_2$) and relative motion ($\mathbf{r} = \mathbf{r}_1 - \mathbf{r}_2$, $\mathbf{p} = \frac{m_2}{M}\mathbf{p}_1 - \frac{m_1}{M}\mathbf{p}_2$), the interaction potential $U$ between the atom and the electromagnetic field follows (except where otherwise noted, hereafter, we use atomic units $e^2 = \hbar = \mu = 1$)

$$
\begin{aligned}
U = \quad & - \quad E_0\left[\sin^2(\frac{\pi t}{NT})\cos(\omega t) + \frac{1}{2N}\sin(\frac{2\pi t}{NT})\sin(\omega t)\right]z \\
& - \quad E_0\frac{\omega}{c}\left[\sin^2(\frac{\pi t}{NT})\sin(\omega t) - \frac{1}{2N}\sin(\frac{2\pi t}{NT})\cos(\omega t)\right](yz + yZ + Yz) \\
& - \quad \frac{E_0}{c}\sin^2(\frac{\pi t}{NT})\cos(\omega t)\left(\frac{yp_z - zp_y}{\tilde{\mu}} + \frac{Yp_z - Zp_y}{\mu} + \frac{yP_z - zP_y}{M}\right),
\end{aligned}
$$

$$(5)$$

where $M = m_1 + m_2$, $\mu = \frac{m_1 m_2}{M}$ and $\tilde{\mu} = \frac{m_1 m_2}{m_2 - m_1}$. After neglecting the term proportional to $1/M$ and using $\mu \approx \tilde{\mu} \approx m_1 = 1$, the interaction potential can be divided into a term dependent only on the relative coordinates

$$
\begin{aligned}
U_1(\mathbf{r},t) = \quad & - \quad E_0\left[\sin^2(\frac{\pi t}{NT})\cos(\omega t) + \frac{1}{2N}\sin(\frac{2\pi t}{NT})\sin(\omega t)\right]z \\
& - \quad E_0\frac{\omega}{c}\left[\sin^2(\frac{\pi t}{NT})\sin(\omega t) - \frac{1}{2N}\sin(\frac{2\pi t}{NT})\cos(\omega t)\right]yz \\
& - \quad \frac{E_0}{c}\sin^2(\frac{\pi t}{NT})\cos(\omega t)\hat{L}_x \ ,
\end{aligned}
$$

$$(6)$$

where $\hat{L}_x = yp_z - zp_y$ is the $x$-component of the angular momentum operator of the electron relative to proton, and the coupling term

$$
\begin{aligned}
U_2(\mathbf{r},\mathbf{R},t) = \quad & - \quad E_0\frac{\omega}{c}\left[\sin^2(\frac{\pi t}{NT})\sin(\omega t) - \frac{1}{2N}\sin(\frac{2\pi t}{NT})\cos(\omega t)\right] \\
& \times \quad (yZ + zY) \\
& - \quad \frac{E_0}{c}\sin^2(\frac{\pi t}{NT})\cos(\omega t)\left(Yp_z - Zp_y\right).
\end{aligned}
$$

$$(7)$$

Thus, the total Hamiltonian of the system turns into

$$H(\mathbf{r},\mathbf{R},t) = \frac{\mathbf{P}^2}{2M} + h_0(\mathbf{r}) + U_1(\mathbf{r},t) + U_2(\mathbf{r},\mathbf{R},t) \ , \tag{8}$$

where $h_0(\mathbf{r}) = \frac{\mathbf{p}^2}{2\mu} - \frac{1}{r}$ shows the Hamiltonian of a unit-charged particle with the reduced mass $\mu$ in the attractive Coulomb field. It is worth noting that, in the dipole approximation, the spatial dependence of the vector field $\mathbf{A}$ is neglected; thus, the magnetic field effect is excluded and the separation of the CM becomes possible, which eventually results in the Hamiltonian of the form $H(\mathbf{r},t) = h_0(\mathbf{r}) + \mathbf{r} \cdot \mathbf{E}(t)$, where $\mathbf{E}(t) = \mathbf{E}(\mathbf{r} = 0, t)$ (see (3)).

For the hydrogen atom $\mathbf{P} = M\mathbf{V} \gg \mu\mathbf{v}$, we can apply the quantum-quasiclassical approach [16,23–26], where the heavy CM is considered to be a classical object and the light electron relative to the proton is treated quantum mechanically. Hence, our problem is reduced to the simultaneous integration of the following system of coupled equations:

$$i\frac{\partial}{\partial t}\psi(\mathbf{r},t) = [h_0(\mathbf{r}) + U_1(\mathbf{r},t) + U_2(\mathbf{r},\mathbf{R},t)]\psi(\mathbf{r},t) \ , \tag{9}$$

$$\frac{d}{dt}\mathbf{P} = -\frac{\partial}{\partial\mathbf{R}}H_{eff}(\mathbf{R}(t),\mathbf{P}(t)) \ , \tag{10}$$

$$\frac{d}{dt}\mathbf{R} = \frac{\partial}{\partial\mathbf{P}}H_{eff}(\mathbf{R}(t),\mathbf{P}(t)) \ , \tag{11}$$

with the effective Hamiltonian

$$H_{eff}(\mathbf{R}, \mathbf{P}) = \frac{\mathbf{P}^2}{2M} + \langle \psi(\mathbf{r}, t) | U_2(\mathbf{r}, \mathbf{R}, t) | \psi(\mathbf{r}, t) \rangle \, , \tag{12}$$

while the initial wave function is supposed to be the hydrogen atom wave function of the ground state $\psi(\mathbf{r}, t = 0) = \phi_{100}(\mathbf{r})$ and the initial conditions for the CM position and momentum are

$$\mathbf{R}(t = 0) = 0 \, , \ \ \mathbf{P}(t = 0) = 0 \, . \tag{13}$$

We integrate the time-dependent three-dimensional Schrödinger Equation (9) by applying the two-dimensional discrete-variable representation method (DVR) [27] and simultaneously integrate the Hamilton equations of motion (10) and (11) with the Störmer-Verlet method [28] adapted in [16,25,26] for the quantum-quasiclassical case:

$$\mathbf{P}(t_n + \frac{\Delta t}{2}) = \mathbf{P}(t_n) - \frac{\Delta t}{2} \frac{\partial}{\partial \mathbf{R}} H_{eff}\left( \mathbf{R}(t_n), \mathbf{P}(t_n + \frac{\Delta t}{2}) \right) ,$$

$$\mathbf{R}(t_n + \Delta t) = \mathbf{R}(t_n) + \frac{\Delta t}{2} \{ \frac{\partial}{\partial \mathbf{P}} H_{eff}\left( \mathbf{R}(t_n), \mathbf{P}(t_n + \frac{\Delta t}{2}) \right)$$

$$+ \frac{\partial}{\partial \mathbf{P}} H_{eff}\left( \mathbf{R}(t_n + \Delta t), \mathbf{P}(t_n + \frac{\Delta t}{2}) \right) \} ,$$

$$\mathbf{P}(t_n + \Delta t) = \mathbf{P}(t_n + \frac{\Delta t}{2}) - \frac{\Delta t}{2} \frac{\partial}{\partial \mathbf{R}} H_{eff}\left( \mathbf{R}(t_n + \Delta t), \mathbf{P}(t_n + \frac{\Delta t}{2}) \right) .$$

$$\tag{14}$$

Once the wave packet $\psi(\mathbf{r}, t)$ and $\mathbf{R}(t)$ and $\mathbf{P}(t)$ of the CM are found during the time interval $0 \leq t \leq T_{out}$ of the laser pulse action, we can calculate the ionization and excitation probabilities [29], and analyze the acceleration of the atom.

### 3. Results and Discussion

*3.1. Excitation and Ionization*

In Figure 1, we demonstrate the calculated dependence on the laser frequency $\omega$ of the population $P_g(\omega)$ of the ground state of the hydrogen atom after its interaction with a linearly polarized laser pulse of intensity $I = 10^{14}$ W/cm$^2$. The results of the calculations of the probabilities of excitation $P_{ex}(\omega)$ and ionization $P_{ion}(\omega)$ of the atom for the same laser frequencies, the intensity, and the pulse duration are also presented in the same figure. Here, the total pulse duration was fixed at $T_{out} = NT = 100\pi$ a.u. $\approx 7.6$ fs, which required an increased number of included optical cycles $N$ by increasing the frequency $\omega$. The populations of the ground state of the atom $P_g(\omega)$ were obtained with the standard procedure of projection at the end of the pulse ($t = T_{out}$) of the calculated electron wave packet $\psi(\mathbf{r}, \omega, t = T_{out})$ onto the ground state $\phi_{100}(\mathbf{r})$ of the unperturbed atom:

$$P_g(\omega) = | \langle \psi | \phi_{100} \rangle |^2 = | \int \psi(\mathbf{r}, \omega, T_{out}) \phi_{100}(\mathbf{r}) d\mathbf{r} |^2 \ . \tag{15}$$

To evaluate the probability of the excitation of an atom by a laser pulse $P_{ex}(\omega) = \sum_{n>1}^{\infty} P_n(\omega)$, we applied the following procedure. The calculation of the populations $P_n(\omega)$ of $2 \leq n \leq 8$ states was carried out in exactly the same way as the population of the ground state (15). To take into account the populations $P_n(\omega)$ of states from $n = 9$ and above, we used the "interpolation" procedure proposed in our previous work [29]. The probability of the ionization of the atom $P_{ion}(\omega)$ was calculated using the formula $P_{ion}(\omega) = 1 - P_g(\omega) - P_{ex}(\omega)$ [29].

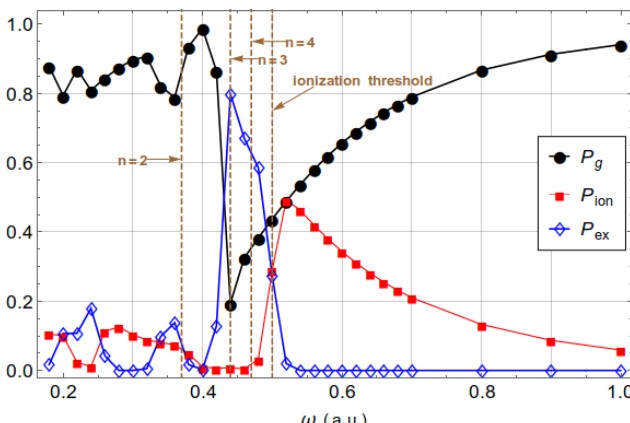

**Figure 1.** The calculated dependences on $\omega$ of the ground-state probability $P_g(\omega)$, and the probabilities of atomic excitation $P_{ex}(\omega)$ and ionization $P_{ion}(\omega)$ for the laser intensity $10^{14}$ W/cm$^2$ and 7.6 fs pulse duration.

The first thing that caught our eye in Figure 1 was the resonant peaks in the probabilities $P_{ex}(\omega)$ near the frequencies $\omega = 0.38$, 0.44, 0.47 (a.u.) defined by the resonant conditions

$$\hbar\omega = \frac{1}{2n^2} - \frac{1}{2n'^2} \tag{16}$$

corresponding to the transitions

$$H_{n=1} + \hbar\omega \rightarrow H_{n'}, \quad n' = 2, 3, 4 . \tag{17}$$

It should be noted that, in the resonance condition (16), we did not take into account the perturbation of the excited state $n'$ due to the dynamic Stark effect, which can give significant corrections with increasing field intensities, especially for highly excited states. The origin of the resonant peaks at $P_{ex}(\omega)$ due to one-photon transitions (17) is clearly confirmed by the calculated time dynamics of the populations $P_n(\omega, t) \xrightarrow{t \rightarrow T_{out}} P_n(\omega)$ of the low-lying states (up to $n = 5$) of the hydrogen atom for some frequencies, including near resonant ones at $\omega = 0.36$ a.u., 0.44 a.u. and 0.48 a.u. (see Figure 2c–e). However, the position of the peak $\omega = 0.24$ a.u. is not described by the resonance condition (16). Nevertheless, the calculated time dynamics of populations $P_n(t)$ shown in Figure 2b clearly demonstrates the dominance of the transition $n = 1 \rightarrow n' = 4$ (17) in the population $P_{ex}(\omega)$ at $\omega = 0.24$ a.u. It is clear that the resonant condition for this transition can be described by the formula

$$2\hbar\omega = \frac{1}{2n^2} - \frac{1}{2n'^2} , \tag{18}$$

with $2\hbar\omega \approx 0.47$ a.u. for $n = 1$ and $n' = 4$. That is, the peak in $P_{ex}(\omega)$ at $\omega = 0.24$ a.u. formed due to a two-photon transition $n = 1 \rightarrow n' = 4$. Some contribution of the state $n' = 3$ in $P_{ex}(\omega)$ at this frequency is also considerable because $2\hbar\omega = 0.48$ a.u. is also close to the resonant frequency of 0.44 a.u. for the transition to the state $n' = 3$. As one can see in Figure 2a, the excitation at the frequency $\omega = 0.22$ a.u. also has a two-photon character due to the resonant transition to $n' = 3$.

It is also noteworthy that the positions of the peaks in $P_{ex}(\omega)$ exactly coincide with the positions of the minima in the population of the ground state $P_g(\omega)$ in the frequency regions, especially in the regions near $\omega \sim (0.22{-}0.24)$ a.u. and $\omega \sim (0.4{-}0.48)$ a.u. where ionization is suppressed. The marked areas of ionization suppression are near two-photon resonant excitation of the states $n' = 3$, 4 and one-photon excitation of the states $n' = 3$, 4 and 5. The suppression of ionization near the one-photon excitation of the state $n' = 2$ at $\omega = 0.38$ a.u. is only slightly noticeable. As the frequency increases and approaches the ionization threshold

$\omega_t = 0.5$ a.u., ionization begins to increase sharply, up to a point $\omega_i = 0.52$ a.u. above the threshold, where $P_{ion}(\omega_i) = P_g(\omega_i)$. Beyond this point, the probability of ionization decreases monotonically with increasing frequency. It is also worth noting that, at the threshold point $\omega_t = 0.5$ a.u, the probability of excitation is equal to the probability of ionization $P_{ex}(\omega_t) = P_{ion}(\omega_t)$.

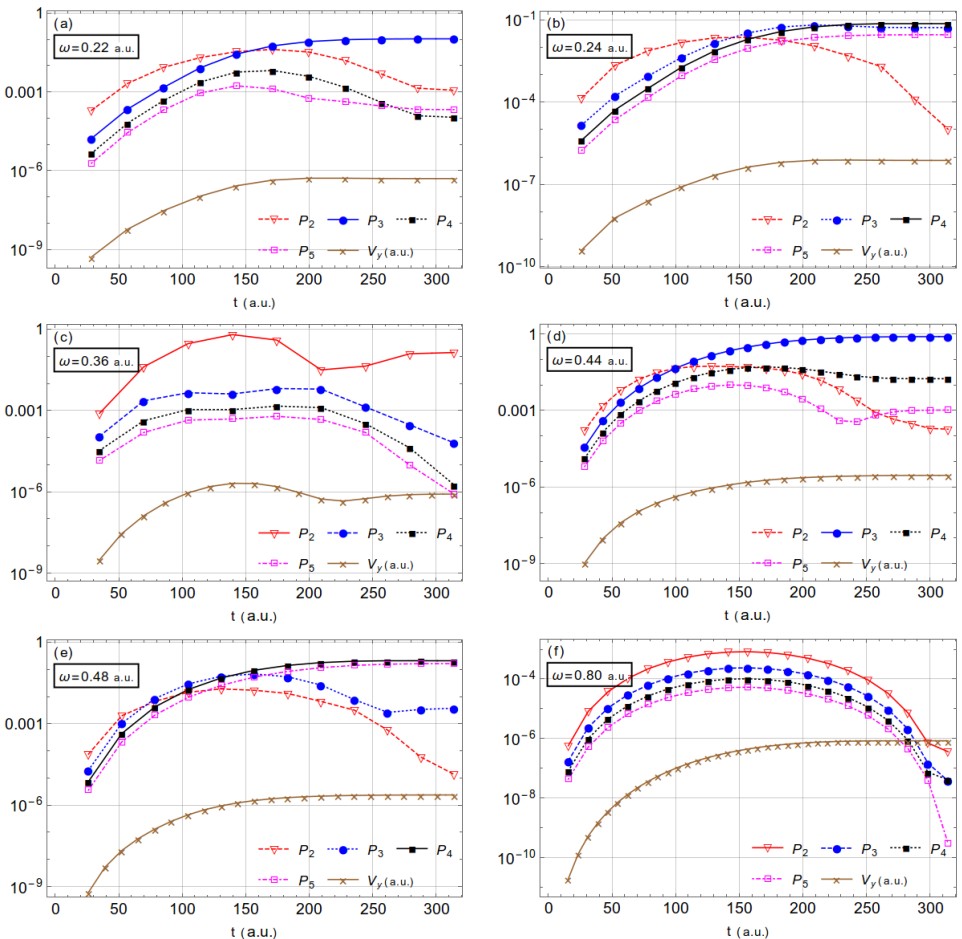

**Figure 2.** The calculated time dynamics of population probabilities $P_n(\omega, t)$ for low-lying states $n = 2$–5 and the atomic CM velocity $V_y(\omega, t)$ in the direction of the pulse spreading. The calculations were performed for frequencies (**a**) $\omega = 0.22$ a.u., (**b**) $\omega = 0.24$ a.u., (**c**) $\omega = 0.36$ a.u., (**d**) $\omega = 0.44$ a.u., (**e**) $\omega = 0.48$ a.u., and (**f**) $\omega = 0.80$ a.u. of a laser field with a $10^{14}$ W/cm$^2$ intensity and 7.6 fs pulse duration.

In Figure 2, we present the calculated time dynamics of the population probabilities of low-lying states for various laser pulse frequencies. The frequencies $\omega = 0.36$ a.u., $\omega = 0.44$ a.u., and $\omega = 0.48$ a.u. in Figure 2c–e correspond to one-photon excitation of atomic levels $n = 2$, 3, 4, and 5 (see also Figure 1), while Figure 2a,b is related to $\omega = 0.22$ a.u. and $\omega = 0.24$ a.u. and illustrates the time dynamics of the population through a two-photon transition. Moreover, in the cases $\omega = 0.22$ a.u., 0.24 a.u., 0.44 a.u., and 0.48 a.u., the processes of excitation of an atom have the two-step character through an intermediate metastable state: the process begins with a transition to the first excited state of the atom $n = 2$, which depopulates rapidly through the transition to higher states due to the resonant interaction of the atom with the laser pulse. This two-step mechanism is most clearly visible in the cases $\omega = 0.22$ a.u. and 0.24 a.u. The resonant population of the lowest excited state $n = 2$, which is observed at the frequency $\omega = 0.36$ a.u. (Figure 2c), is a one-step process and occurs directly without any transitions to an intermediate state. Figure 2f, $\omega = 0.8$ a.u., illustrates a nonresonant case of atomic excitation and relates to

the excitation of the atom above the ionization threshold. It can be seen that, in the latter case, all populated low-lying levels of the atom, with the exception of $n = 2$, are metastable. They are depopulated at the end of the laser pulse, after which only a small part of atoms remains in the excited $n = 2$ state.

### 3.2. Acceleration of Neutral Atoms

Figure 2 also shows the results of calculating the time dynamics of the acceleration of a neutral hydrogen atom by a laser pulse of various frequencies. Here, we present the CM velocities of an atom in the direction of propagation of the laser pulse $V_y(\omega, t)$ calculated as a function of time. It should be noted that, in all cases considered, with the exception of $\omega = 0.8$ a.u., the acceleration of the atom CM repeated the time dynamics of the population of the most populated level. In the case of the above-threshold ionization with $\omega = 0.8$ a.u., the velocity of the CM monotonically increased with time, reaching a maximum at the end of the pulse ($t = T_{out} = 100\pi$ a.u.), while the populations of all low-lying atomic levels grew to the point of maximum intensity of the laser pulse (at the point $t = T_{out}/2$) and began to depopulate after its passage. At the end of the laser pulse, only a small proportion of the atoms in the $n = 2$ state remained in the excited state.

The correlation between the time dynamics of the CM velocity of the atom $V_y(\omega, t)$ and the population probability $P_n(t)$ of the most populated level demonstrated in Figure 2 confirms that the mechanism of acceleration of an atom by a laser pulse is the acceleration of a spatially inhomogeneous electron cloud in excited states due to ponderomotive forces for a frequency below the ionization threshold. In the region of laser frequencies exceeding the ionization threshold (see case $\omega = 0.8$ a.u.), we observed the acceleration of the atom's CM even after passing the critical point $t > T_{out}/2$, when the atom was depopulated. The growth in this region $V_y(\omega = 0.8, t)$ occurred due to ionized electrons, the value of which is significant here (see Figure 1).

In fact, as presented in Figure 3, the calculated dependencies on the laser frequency of the total probability of excitation and ionization of the atom $P_{ex}(\omega) + P_{ion}(\omega)$ and the momentum $P_y(\omega) = MV_y(\omega)$ of its accelerated CM demonstrate their strong correlation with each other. This is a clear demonstration of what is the root cause of the acceleration of the CM of an atom due to a laser field. This is the generation of a nonzero dipole moment between the proton and electron cloud that, under the action of an electromagnetic pulse, has transferred either to the excited state of the atom or to its continuum. Thus, we see that, in the case of the noticeable ionization of an atom, in addition to the acceleration of the neutral atom itself as a whole, the acceleration of the electron falling into the continuous spectrum also contributes to the acceleration of the atom CM. Naturally, for practical purposes of accelerating neutral atoms, one should use frequency regions of laser radiation in which ionization is suppressed compared to the excitation of the atom. In this regard, the frequencies near the two-photon resonances ($n = 3, 4$) $\omega \sim (0.22–0.24)$a.u and one-photon resonances ($n = 3–5$) $\omega \sim (0.42–0.48)$ a.u. (see Figure 1) are promising.

In Table 1 and Figure 4, we present the calculated velocities $V_y(\omega, I)$ of the atom CM as a function of the radiation intensity $I$ for resonant $\omega = 0.24$ a.u., 0.48 a.u., and nonresonant $\omega = 0.8$ a.u. frequencies. It is worth noting the linear dependence of the calculated values $V_y(\omega, I)$ on $I$ for all given frequencies on the double-logarithmic scale in Figure 4. Moreover, the slopes of the calculated curves, with the exception of the frequency $\omega = 0.24$ a.u., were almost the same on the double-logarithmic scale, which corresponds to the linear dependence of the the CM velocity on the intensity: $V_y(\omega, I) \propto I$ (see also Table 1). However, the angle of inclination of the curve $\omega = 0.24$ a.u. increased, which, in this case, precisely gives the quadratic dependence of the velocity of the atomic CM in the considered range of intensities: $V_y(\omega = 0.24, I) \propto I^2$. The physical interpretation of the discovered effect is the following: The cases of acceleration of the CM of the atom considered here, except for $\omega = 0.24$ a.u., correspond to the one-photon mechanism discussed above. In the one-photon case, the effect of accelerating the CM to velocity $V_y(\omega, I)$ is proportional to the photon number density in the laser pulse (i.e., intensity $I$), which is clearly demonstrated

in Figure 4. In the case of $\omega = 0.24$ a.u., the process has a two-photon origin, in which the acceleration of the CM to a velocity $V_y(\omega = 0.24, I)$ is proportional to the square of the photon number density in the laser pulse, which leads to a quadratic dependence $V_y(\omega = 0.24, I) \propto I^2$, as observed in Figure 4.

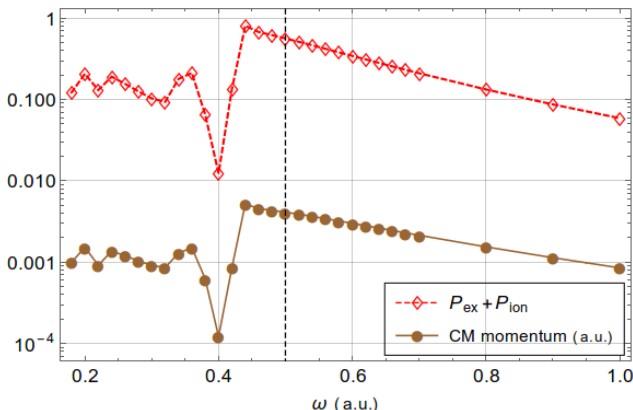

**Figure 3.** The calculated dependence on the laser frequency of the total probability for atomic excitation and ionization $P_{ex}(\omega) + P_{ion}(\omega)$ together with the momentum $P_y(\omega) = MV_y(\omega)$ of the atomic CM at the end of the pulse duration $t = T_{out} = 7.6$ fs. The calculations were performed for a laser intensity of $10^{14}$ W/cm$^2$. The vertical black dashed line indicates the ionization threshold of the atom at $\omega_t = 0.5$ a.u.

It should be noted that the straight line $V_y = 1.96 \times 10^{-12} I$ satisfactorily describes all the calculated points $V_y(\omega = 0.8, I)$ in the entire range of $I$ variation for the nonresonant one-photon case $\omega = 0.8$ a.u. in Figure 4. In the resonant one-photon case $\omega = 0.48$ a.u., we observed a noticeable deviation of the calculated points $V_y(\omega = 0.48, I)$ from the straight line $V_y = 7.53 \times 10^{-12} I$ for laser intensities $I \gtrsim 0.7 \times 10^{14}$ W/cm$^2$. In the case of two-photon resonance $\omega = 0.24$ a.u., a noticeable deviation of the calculated values $V_y(\omega = 0.24, I)$ from the parabola $V_y = 2.05 \times 10^{-26} I^2$ was observed only for $I \gtrsim 1 \times 10^{14}$ W/cm$^2$. This deviation from the linear $\propto I$ and quadratic $\propto I^2$ dependencies with increasing laser intensity in resonant cases may be a consequence of the manifestation of the dynamic Stark effect with increasing intensity, leading to a change in resonance conditions. This hypothesis is supported by the fact that, in the nonresonant case, there was practically no deviation of the calculated velocities from the linear dependence in the entire considered intensity range. A complete depopulation of the ground state into excited states and a continuum with increasing laser intensity can also prevent the acceleration of the atom.

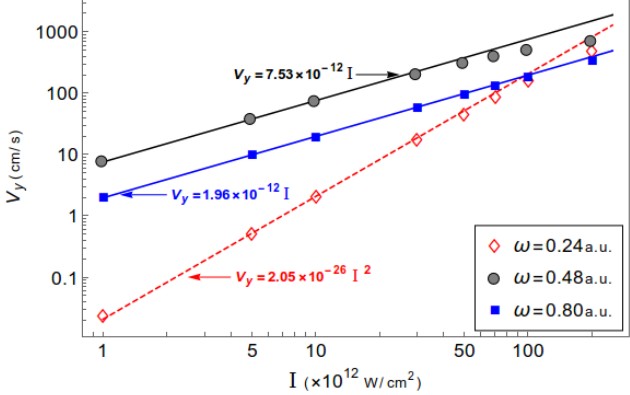

**Figure 4.** The calculated CM velocity in the y-direction as a function of intensity $I$ for $\omega = 0.24$ a.u. (empty red diamonds), 0.48 a.u. (full black circles), 0.8 a.u. (full blue squares), and a 7.6 fs pulse duration.

**Table 1.** The calculated CM velocity for different laser intensities $I$ and frequencies $\omega = 0.80$ a.u., 0.48 a.u., 0.24 a.u., and a 7.6 fs pulse duration.

| $I\ (\times 10^{12}\ \text{W/cm}^2)$ | $V_y$ (cm/s) | | |
|---|---|---|---|
| | $\omega = 0.80$ a.u. | $\omega = 0.48$ a.u. | $\omega = 0.24$ a.u. |
| 1 | 1.97 | 7.90 | 0.024 |
| 10 | 19.6 | 75.7 | 2.06 |
| 50 | 95.2 | 317 | 46.4 |
| 100 | 184 | 517 | 163 |
| 200 | 344 | 722 | 488 |

*3.3. Influence of Nondipole Effects on Excitation and Ionization Processes*

We also analyzed the influence on the excitation and ionization processes of the nondipole effects. Table 2 illustrates the deviation of $P_g(\omega)$ and $P_{ex}(\omega)$ calculated for the dipole and nondipole approaches. We see that the influence of nondipole effects in the considered range of frequencies on the excitation and ionization of the atom was insignificant. The relative deviation did not exceed the value $0.5 \times 10^{-3}$.

**Table 2.** The probabilities of the population of the ground state $P_g$ and total excitation $P_{ex}$ calculated for the dipole and nondipole approaches for a few laser frequencies with intensity $10^{14}$ W/cm$^2$ and a 7.6 fs pulse duration.

| $\omega$ | $P_g$ | | | $P_{ex}$ | | |
|---|---|---|---|---|---|---|
| | Dipole | Nondipole | $|\Delta P|$ [1] | Dipole | Nondipole | $|\Delta P|$ [1] |
| 0.30 | 0.896815 | 0.896805 | $1.05 \times 10^{-5}$ | $6.3410 \times 10^{-5}$ | $6.3414 \times 10^{-5}$ | $6.59 \times 10^{-5}$ |
| 0.40 | 0.987600 | 0.987599 | $1.34 \times 10^{-6}$ | $2.3058 \times 10^{-3}$ | $2.3052 \times 10^{-3}$ | $2.51 \times 10^{-4}$ |
| 0.48 | 0.382308 | 0.382294 | $3.54 \times 10^{-5}$ | $5.8582 \times 10^{-1}$ | $5.8568 \times 10^{-1}$ | $2.37 \times 10^{-4}$ |
| 0.52 | 0.488483 | 0.488465 | $3.74 \times 10^{-5}$ | $1.9974 \times 10^{-2}$ | $1.9984 \times 10^{-2}$ | $4.84 \times 10^{-4}$ |
| 0.80 | 0.867150 | 0.867131 | $2.20 \times 10^{-5}$ | $4.6748 \times 10^{-7}$ | $4.6756 \times 10^{-7}$ | $1.64 \times 10^{-4}$ |
| 1.00 | 0.941058 | 0.941045 | $1.39 \times 10^{-5}$ | $4.8210 \times 10^{-7}$ | $4.8218 \times 10^{-7}$ | $1.68 \times 10^{-4}$ |

[1] $|\Delta P| = \left| \frac{P_{Dipole} - P_{Nondipole}}{P_{Dipole}} \right|$.

## 4. Conclusions

We investigated the acceleration of a hydrogen atom in strong ($10^{12} - 2 \times 10^{14}$ W/cm$^2$) linearly polarized short-wavelength (5 eV $\lesssim \hbar\omega \lesssim 27$ eV) electromagnetic pulses of about 8 fs duration. The study was carried out within the framework of the hybrid quantum-quasiclassical approach, in which the coupled time-dependent Schrödinger equation for an electron and the classical Hamilton equations for the atom CM were simultaneously integrated. It was found that the origin of atomic acceleration is the transition of a part of the electron cloud, under the action of electromagnetic wave, from the ground to excited states.

The optimal conditions with respect to the frequency and intensity of the electromagnetic pulse for the acceleration of atoms without their noticeable ionization were found in the analyzed region. We showed that the regions with the most potential in this regard are the regions of one-photon resonances $n = 3$–$5$ ($\omega = (0.4 - 0.48)$ a.u.) and two-photon resonances $n = 3, 4$ ($\omega = (0.22 - 0.24)$ a.u.). We demonstrated that, for an intensity of $I = 2 \times 10^{14}$ W/cm$^2$, the hydrogen atom is maximally accelerated at $\omega = 0.48$ a.u. to a velocity of $V_y \simeq 700$ cm/s. At a frequency of $\omega = 0.24$ a.u., the acceleration is noticeably less than this value and reaches 500 cm/s; however, due to the two-photon mechanism of excitation of the atom at this frequency, a further increase in intensity should lead to a more significant acceleration than at $\omega = 0.48$ a.u. due to the established dependence for the two-photon resonance $V_y \propto I^2$. In this regard, it is also interesting to consider the three-photon excitation mechanism due to the supposed dependence $V \propto I^3$ for the acceleration of atoms in this case. However, it is clear that this effect starts to be significant at rather high laser intensities. Nevertheless, it looks promising to investigate the region of three-photon resonances for higher intensities due to the cubic dependence on the intensity of the CM velocity here; however, this demands special consideration.

It appears to us that, among the current tasks of further research into the possibility of accelerating atoms using electromagnetic radiation, the inclusion of the spatial inhomogeneity of a laser beam in the computational scheme and its generalization to helium and neon atoms are feasible next steps, where experimental data are more attainable [1,15,17]. It is also worth considering in our approach the influence of laser radiation polarization on the acceleration of an atom and the possibility of obtaining twisted atoms in this case.

**Author Contributions:** Conceptualization, supervision, V.S.M.; investigation, writing, V.S.M. and S.S. All authors have read and agreed to the published version of the manuscript.

**Funding:** This work was supported by the Russian Science Foundation under Grant No. 20-11-20257.

**Data Availability Statement:** The data presented in this study are available on request from the corresponding author.

**Conflicts of Interest:** The authors declare no conflicts of interest.

## Abbreviations

The following abbreviations are used in this manuscript:

CM      center-of-mass
DVR     discrete-variable representation

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
