# Peer review of "Acceleration of Neutral Atoms by Strong Short-Wavelength Short-Range Electromagnetic Pulses"

_photonics, doi:10.3390/photonics10121290_

Round 1

Reviewer 1 Report

Comments and Suggestions for Authors

This paper studies the acceleration of neutral hydrogen atoms subject to femtosecond laser fields as a function of photon energy and inetnsity using a semiclassical approach where the electronic motion is quantum and a classical treatment of the center-of-mass motion. The treatment also allows them to understand the role of non-dipole effects which where subsequently shown to be not important over the whole range of energies considered. The paper clearly outlines past work and also the approach used in this study. They show that their calculations produce sensible results in terms of excitation as a function of wavelength. They then present the final velocity as a function of photon energy and intensity and identify optimum parameters for acceleration. This is a interesting and useful paper that also points towards further work that would take into account non-uniform intensity profiles, other atoms and polarization schemes which could be linked more closely with experimental studies.

Comments on the Quality of English Language

The language is mostly clear with a coherent and well ordered discussion of the problem and its results. There are a few typos.

Author Response

We would like to thank the referee for his/her thorough review of our work as well as given remarks.

Below we address all his/her comments

Comment : The language is mostly clear with a coherent and well ordered discussion of the problem and its results. There are a few typos.

Response: Following the referee's recommendation the found typos have been undertaken throughout the manuscript.

Reviewer 2 Report

Comments and Suggestions for Authors

The paper studies the acceleration of neutral atoms by strong short-wavelength short-range electromagnetic pulses. Their theory is based on a hybrid quantum-quasiclassical approach in which the coupled time-dependent Schrödinger equation for an electron and the classical Hamilton equations for the CM of an atom are simultaneously integrated. In general, this paper is well presented and clearly written. My main concern is on the relevance of the present model with the experimental measurement. In Ref. 1, it has been shown that acceleration of neutral atoms mainly comes from the ponderomotive force during the laser pulse owing to the intensity gradient in the focused laser beam. The final velocity of the atom is closely related to the Rayleigh length and beam waist.  However,the laser focus effects have been completely neglected, as can be seen in Eq. (1).  I think the authors should address more clearly the validity of the present theoretical model in explaining the acceleration of neutral atoms by laser fields.

In addition, in Fig.3 the lines show a clear minimum at photon energy 0.4 a.u. Can they give some physical explanations for this feature?  

Author Response

We would like to thank the referee for his/her thorough review of our work and its positive evaluation, as well as his/her recommendations and comments.

Below we address all his/her questions/comments

Comment : My main concern is on the relevance of the present model with the experimental measurement. In Ref. 1, it has been shown that acceleration of neutral atoms mainly comes from the ponderomotive force during the laser pulse owing to the intensity gradient in the focused laser beam. The final velocity of the atom is closely related to the Rayleigh length and beam waist.  However,the laser focus effects have been completely neglected, as can be seen in Eq. (1).  I think the authors should address more clearly the validity of the present theoretical model in explaining the acceleration of neutral atoms by laser fields.

Response: We have considered short-wave pulses with a periodic intensity gradient determined by the inhomogeneity ~ kz=ωz/c=z/(λ/2π) of the electromagnetic wave, i.e. a scale on the order of the laser wavelength (λ/2π~35-7) nm. Since it is at least two orders of magnitude narrower than the 17500 nm laser waist in experimental work [1], we believe that our approach neglecting the effect of laser focusing is reasonable. In the case of a Ti:Sa laser (conditions of work [1]), the scale of the inhomogeneity of the electromagnetic wave λ~(700-1100) nm approaches the value of the laser waist, and taking it into account becomes a significant task, as well as generalizing the developed approach for a two-electron atom. Our intention to include the non-uniform intensity profile in a future work is already mentioned in the last paragraph of the sectionConclusions” (lines 273-275).

Comment : In addition, in Fig.3 the lines show a clear minimum at photon energy 0.4 a.u. Can they give some physical explanations for this feature?

Response: As we explain in our work, the peaks at points ω=0.36a.u. and ω=0.44a.u. are of a resonant nature. And the most distant point from these resonances that lies between them is point ωmin=0.4a.u., which lies exactly in the middle. Since it is farthest from both resonances, a minimum is observed in it.

Reviewer 3 Report

Comments and Suggestions for Authors

1. Please add more motivations about this research. What would be affected by this research? For example, the determination and precision about the laser energy?

2. Please explain why the laser energy is picked at 10^14 and 7.6fs. What if replace the hydrogen with other atoms?

3. Please consider to generate figures to clearly compare single-photon and two photon. That would be another interest point from the readers except from the frequency dependency.

Author Response

We would like to thank the referee for his/her thorough review of our work, as well as his/her questions and comments.

Below we address all his/her questions/comments

Comment 1 : Please add more motivations about this research. What would be affected by this research? For example, the determination and precision about the laser energy?

Response: If we understand the referee's question correctly, it refers to the laser frequency (photon energy). Indeed, in some devices (for example, XFEL), it is problematic to accurately fix the laser frequency. In this regard, our calculations of the frequency dependencies (with rather small step by laser frequency) for the acceleration of an atom, for its ionization and excitation can be useful for choosing optimal conditions when planning experiments. We have added the appropriate sentence in t Introduction (lines 50-53).

Comment 2 : Please explain why the laser energy is picked at 10^14 and 7.6fs. What if replace the hydrogen with other atoms?

Response: The study of atoms in strong laser fields in the intensity range of about 1014W/cm2 and duration of the order of a few tens fs is currently the subject of intensive experimental research (see, for example [7,17, D. Trabert et. al. PRL 127, 273201 (2021)] and refs. therein). The region ~ 1014W/cm2 and ~ several fs is also typical for theoretical research (see, for example [8,13,15,Yuan, K.J.; et al, Phys. Rev. A 101, 023411 (2020), Pauly, T.; et al, Phys. Rev. A 102, 013116 (2020)] ). Moreover, in the experimental investigations there is a clear tendency to move into the region of shorter pulses than a few tens fs. All this stimulated our interest in chosen intensity range, and the value Tout = N 2π/ω=100π a.u. = 7.6 fs (N is the number of optical cycles) was chosen as the pulse duration. We have added correspondent discussion in Introduction, lines 46-50.

Possible generalization of the developed approach for other atoms is mentioned at the last paragraph of the conclusion (lines 273-276). Mention here also the comments from the Referee 1 report:``...useful paper that also points towards further work that would take into account non-uniform intensity profile, other atoms …. which could be linked more closely with experimental studies”.

Comment 3 :Please consider to generate figures to clearly compare single-photon and two photon. That would be another interest point from the readers except from the frequency dependency.

Response: Following this advice, we added Table 1 to the text with numerical values of atomic velocities at the end of the electromagnetic pulse depending on its intensity and frequencies for one-photon and two-photon mechanisms. From these data, constants A(ω=0.8)=1.96x10-12, A(ω=0.48)=7.53x10-12 and B(ω=0.24)= 2.05x10-26 were extracted for the curves Vy=A I and Vy=B I2 approximating the one-photon and two-photon cases, respectively (shown in Fig.4). Corresponding discussion is added at the end of subsection 3.2 (lines 228-239).

Reviewer 4 Report

Comments and Suggestions for Authors

Referee report

The manuscript by V. Melezhik and S. Shadmehri  is devoted to the theoretical investigation of the problem of acceleration of an atom by a strong electromagnetic pulse due to the non-dipole atom-laser interaction. As an example the non-dipole effects in interaction of a laser with a hydrogen atom is  considered. For calculating the acceleration as well as the probabilities of excitation and ionization of the atom the previously developed quantum-classical approach is used in which the motion of the electron is considered quantum-mechanically while the center-of-mass motion is considered classically. The paper is clearly written, the results are well presented and look reliable. They may be useful for further experimental investigations of non-dipole effects in laser-atom interaction. I recommend to publish this paper as it is.

Author Response

We would like to thank the referee for his/her thorough review of our work and recommendation to publish it as it is.

Round 2

Reviewer 2 Report

Comments and Suggestions for Authors

I agree the acceptance of this paper.